# From traits to puffs: The interplay of personality, pandemic stress, and smoking behaviors

Ying Tian[1], Weiyi Xiang[1], Silvia Dzhugaryan[1‡], Dayoung Bae[1,2],
Jessica Barrington-Trimis[1,2], Terry Church[1,3]*

1 University of Southern California Keck School of Medicine, Institution of Addiction Science, Los Angeles, California, United States of America, 2 Department of Population and Public Health Sciences, University of Southern California, Keck School of Medicine, Los Angeles, California, United States of America, 3 University of Southern California, Alfred E. Mann, School of Pharmacy and Pharmaceutical Sciences, Los Angeles, California, United States of America

☯ These authors contributed equally to this work.
‡ These authors also contributed equally to this work.
* tdchurch@usc.edu

## Abstract

Smoking, a leading cause of chronic diseases, is often used to cope with stress, which has been heightened by the pandemic due to health and economic concerns. Studies have shown that the Big Five personality traits are linked to smoking behavior, suggesting that different personality traits influence nicotine use in varying ways. However, there remains a significant gap in understanding how individuals with different personalities respond to nicotine use under stress. This study aims to investigate how nicotine dependence changes for different Big Five personalities under the pandemic stress and whether other stress-related factors influence nicotine dependence during COVID-19. This cross-sectional study collect data from randomly selected adults aged 18−30 in the US. The Big Five Personality Model assessed personality traits, and nicotine dependence was measured with the Hooked-on Nicotine Checklist. Stress was evaluated using the Perceived Stress Scale, while demographics and other pandemic-related stressors were gathered through structured questions. Correlation and multiple logistic regression models were used for data analysis. The main findings showed that both before (r=−.25, p<.001) and during (r=−.19, p<.001) the pandemic, agreeableness was significantly negatively associated with nicotine dependence, indicating that higher agreeableness was linked to lower nicotine dependence. Similarly, conscientiousness was negatively correlated with nicotine dependence both before (r=−.123, p<.001) and during COVID-19 (r=−.19, p<.001). Although no direct association was found between perceived stress, personality traits, and smoking behavior, the analysis identified that external stressors played a moderating role. These findings emphasize the importance of understanding how different personality traits influence young people's dependence on nicotine under stress. The outcome can guide the design of targeted nicotine withdrawal interventions and inform effective public health strategies.

**Data availability statement:** Data cannot be shared publicly because of consent agreement. Data are available from the University of Southern California Institutional Review Board for researchers who meet the criteria for access to confidential data. Name: Brianne Mongeon Email: hrpp@usc.edu. Title/Position: Senior Analyst, USC Human Research Protection Program

**Funding:** The Institute for Addiction Science provided Ying Tian, Weiyi Xiang, and Silvia Dzhugaryan a total of $500 to recruit and compensate participants. The grant number is not available.

**Competing interests:** The authors have declared that no competing interests exist.

## Introduction

Tobacco use has been one of the most serious public health problems. Studies have shown that tobacco use can cause cardiovascular damage, respiratory conditions like emphysema and chronic obstructive pulmonary disease (COPD), and mouth cancer [1]. By 2020, 22.3% of the global population uses tobacco, and nearly 8 million people die from tobacco use each year [2]. However, among the fatal diseases, tobacco-related diseases can be effectively reduced or even artificially avoided, and reducing tobacco use can effectively reduce the global burden of disease [3]. Therefore, it is important to understand the motivations and mechanisms underlying smoking behavior and nicotine dependence.

Personality traits have been shown to play an important role in smoking behavior, and there are significant differences in smoking behavior among people with different personalities. Studies have found that people with high levels of extraversion and neuroticism are more likely to start smoking. This may be due to their impulsivity, social tendencies, or emotional instability [4,5]. On the other hand, individuals who score high in conscientiousness and agreeableness tend to smoke less. These traits are often linked to better self-control and respect for social norms [4]. Openness, however, is associated with higher nicotine dependence. This suggests that people who are more open to experiences may be more willing to experiment with substances like tobacco [5]. Different personality traits not only influence the individual's propensity to smoke but may also determine their level of nicotine dependence and ease of quitting. For men, those with lower neuroticism and extraversion and higher openness were more likely to be successful in quitting, while women with lower conscientiousness were more likely to quit [6]. These findings suggest that individual personality traits may play a key role in the development and maintenance of smoking behavior.

Moreover, stress is also an important factor that influences smoking behavior. People often smoke as a means of coping with negative emotions and to relieve major stressful events in their lives [7]. Studies have shown that when smokers' stress levels or negative emotions increase, their daily tobacco consumption, time spent smoking, and frequency of smoking also increase significantly [8–10]. This phenomenon may be due to the transient relaxing effect of the nicotine component of cigarettes. By influencing the central nervous system, nicotine controls the production of dopamine, resulting in a transient pleasure experience and a calming effect [11]. High levels of stress are also one of the main reasons why smokers relapse. Studies have shown that stress significantly increases the likelihood of relapse by making smokers more inclined to remember the environment associated with smoking and enhancing their memory bias for smoking-related items [12].

In the specific context of the COVID-19 pandemic, the complexity of smoking behavior has been further exacerbated by the unprecedented amplification of psychological stress. Surveys have shown that the COVID-19 epidemic has disrupted people's daily, increased anxiety of isolation, unemployment, and health anxiety [13,14]. One study noted that during the COVID-19 pandemic, 28% of participants (n = 3396) reported an increase in their smoking, with increased stress, prolonged

time spent at home, and boredom cited as the main reasons [15]. In addition, 16.8% of young nicotine e-cigarette users in 2020 reported that they had increased their smoking frequency due to epidemic-related stressors [16].

Although a large number of studies have validated personality traits and stress as a cause of smoking, their interaction has rarely been noted. Thus, the objective of this study is to investigate how stress from the COVID-19 pandemic influenced nicotine dependence based on the Big Five personality traits (e.g., neuroticism, extraversion, openness, agreeableness, and conscientiousness) within nicotine users vs non-nicotine users in young adults. We have three hypotheses: [1] Personality is related to nicotine. [2] people with neuroticism and extraversion personalities show an increase in nicotine dependence, whereas people with conscientious personalities do not show significant changes or decreases in nicotine dependence. [3] Other stress-related factors influence nicotine dependence during COVID-19, such as health concerns, access to nicotine, economic factors, mental health, routine disruption, social networking, and traumatic events.

## Methods

This study has been approved and exempted by the University of Southern California Institutional Review Board and participant authorization was obtained in the form of posters and questionnaires (IRB Study ID: UP-23–01234). Our observational cross-sectional survey was conducted from February 2024 to May 13, 2024. Participants aged 18–30, able to provide informed consent, were recruited through flyers at the University of Southern California, social media platforms (Instagram and Snapchat), and Amazon Mechanical Turk (MTurk). Incentives included an Apple AirPod, eight Apple AirTags, and a > 5% chance to win a $10 Amazon gift card. Of the 433 respondents, < 30% of data were excluded due to ineligibility or incomplete responses, yielding a final sample of 323 participants. Inclusion criteria required participants to be aged 18–30 and able to complete the survey. Participants who have used any nicotine product in their lifetime answered the following questions measuring sociodemographics, personality (Big Five) if they used any nicotine products, what type of nicotine product, nicotine consumption hooked on nicotine checklist (HONC), other influences of nicotine behavior, and retrospectively reported perceived stress (PSS) levels both before COVID-19 (2018–2019) and during COVID-19 (2020–2021). Non-nicotine users only answered the following questions: measuring, sociodemographics, personality (Big Five), and whether they have ever used any nicotine products. All measurements had the same variables and scoring in conjunction with the following timeline.

### Measures

Personality traits were measured using the 44-item Big Five Personality Inventory (Goldberg, 1993), which assesses five dimensions: Extraversion, Neuroticism, Agreeableness, Conscientiousness, and Openness. Items were rated on a 5-point Likert scale ranging from 1 = "Strongly Disagree" to 5 = "Strongly Agree." Some items required reverse coding, and final scores were calculated by summing all items within each dimension, with higher scores indicating stronger traits. This scale has demonstrated high reliability (Cronbach's α = 0.70–0.90).

Nicotine dependence was assessed using the Hooked on Nicotine Checklist (HONC), a 10-item tool designed to evaluate the loss of autonomy over nicotine use. Responses were scored as 1 = "Yes" and 0 = "No," with total scores ranging from 0 to 10. Dependence levels were categorized as follows: 0 = No dependence, 1–2 = Low dependence, 3–5 = Moderate dependence, 6–8 = High dependence, and 9–10 = Severe dependence. Participants who reported lifetime nicotine use (yes/no) were further queried about specific products used, including e-cigarettes, cigars, hookah, smokeless tobacco, non-combustibles, and dissolvables. Only nicotine users completed the HONC and related measures of nicotine behaviors.

Stress levels were measured using the Perceived Stress Scale (PSS), a 10-item questionnaire assessing perceived stress in the past month. Items were rated on a 5-point Likert scale from 0 = "Never" to 4 = "Very Often." Scores were calculated by summing all item responses, with higher scores indicating greater stress. Several items required reverse coding before summing. The PSS has demonstrated high reliability (Cronbach's α = 0.70–0.90).

 

To further explore the relationship between stress regulation, personality, and nicotine use behavior, an additional stress-related factors questionnaire was administered. This measure assessed the influence of factors such as stress relief, relaxation, boredom, social situations, and academic pressure on nicotine use. Participants rated these factors on a 5-point Likert scale ranging from 1 = "Not at all" to 5 = "Extremely," with higher scores indicating a stronger influence. This questionnaire, completed only by nicotine users, aimed to identify specific stressors contributing to nicotine consumption and demonstrated high internal consistency (Cronbach's α = 0.92). Non-nicotine users were excluded from this assessment.

Demographic information, including age, gender, race/ethnicity, financial status, and health conditions, was collected to contextualize findings and assess sociodemographic influences on nicotine use. Participants provided responses regarding their age, assigned sex at birth, gender identity, race/ethnicity, and personal financial situation, alongside self-reported health conditions diagnosed by a healthcare provider. Together, these measures provided a detailed understanding of how stress influences the relationship between personality and nicotine use.

### Data analyze

This study used SPSS for descriptive statistics, correlations, and multiple regression models to ensure the accuracy of experimental data analysis and examine relationships between key variables.

First, descriptive statistics were used to summarize and describe participants' demographic characteristics, health status, and economic status to provide a clear base profile for subsequent analyses. The 'Crosstabs' function was used in SPSS to compare nicotine users and non-users, generating contingency tables that provide an overview of participant diversity.

Subsequently, correlations were conducted to test the hypothesis of the moderate effect of pandemic stress on the relationship between personality traits and nicotine dependence. In SPSS, the 'Pearson correlation' function assessed the strength and direction of associations between variables, providing a reliable measure of their linear relationships.

Moreover, multiple logistic regression analyses were performed using the built-in "Logistic Regression" tool in SPSS to effectively assess the pathways of nicotine dependence across personality traits and pandemic-related stress. This approach explores the moderate effects of COVID-related concerns between personality traits and nicotine dependences.

## Results

### Descriptive results, by nicotine use status

Demographic data were collected on age, sex at birth, race/ethnicity, sexual orientation, gender identity, diagnosed health diseases, and financial income as "Table 1". The study includes 324 participants, with 269 nicotine users and 54 non-nicotine users. The average age of all participants is 26.10 years; nicotine users averaged 26.78 years, while non-nicotine users averaged 22.69 years. There are more males and fewer females among the nicotine users (57.4% male and 33.1% female) than the non-nicotine users (33.3% male and 66.9% female). Nicotine users have a lower percentage of Heterosexual/Straight individuals (63.6%), and a higher percentage of Bisexual individuals (30.5%) compared to non-nicotine users (81.5% and 11.1%, respectively). In terms of health diagnoses, there are, more nicotine users were diagnosed with a disease (66.5%) compared to non-nicotine users (16.7%), including 28.6% of nicotine users with a physical disability. Financially, more nicotine users lived comfortably (54.6%) compared to non-nicotine users (42.6%). A higher percentage of non-nicotine users met their needs with little left over (33.3% vs. 26.0%) or just met basic expenses (20.4% vs. 19.0%).

As shown in "Table 2", nicotine users had higher extraversion scores (25.73 vs. 23.96, p < .05) and lower agreeableness (29.23 vs. 32.85, p < .001) and conscientiousness scores (29.03 vs. 31.25, p < .001) compared to non-nicotine users. For nicotine users, we also assessed HONC and PSS scores before and during COVID-19. The average HONC score slightly decreased during COVID-19 (4.71, SD = 4.00) compared to before (4.85, SD = 3.96). PSS scores were stable, with an average of 26.77 (SD = 3.50) during COVID-19 and 26.62 (SD = 3.63) before COVID-19.

**Table 1. Descriptive statistics for participants, demographic information, and the P (chi-square) value between nicotine users (n = 269) and non-nicotine users (n = 54).**

| Variable | Nicotine users | Non-nicotine users | P (chi-square) |
|---|---|---|---|
| Mean Age | 26.78 | 22.69 | <.001** |
| **Sex at birth** | | | <.001** |
| Male | 170 (57.4%) | 18 (33.3%) | |
| Female | 98 (33.1%) | 36 (66.9%) | |
| Intersex | 1 (0.3%) | 0 (0.0%) | |
| **Race/ethnicity** | | | <.001** |
| Asian or Pacific Islander | 47 (17.5%) | 40 (74.1%) | |
| Black or African American | 6 (2.2%) | 1 (1.9%) | |
| Hispanic or Latino | 8 (3.0%) | 6 (11.1%) | |
| Native American/Alaskan Native | 9 (3.3%) | 0 (0.0%) | |
| White or Caucasian | 192 (71.4%) | 6 (11.1%) | |
| Middle Eastern | 5 (1.9%) | 0 (0.0%) | |
| Another (not listed) | 2 (0.7%) | 1 (1.9%) | |
| **Sexual identity** | | | <.05* |
| Heterosexual/Straight | 171 (63.6%) | 44 (81.5%) | |
| Homosexual | 13 (4.8%) | 3 (5.6%) | |
| Bisexual | 82 (30.5%) | 6 (11.1%) | |
| Pansexual | 1 (0.4%) | 0 (0.0%) | |
| Asexual | 1 (0.4%) | 0 (0.0%) | |
| Another (not listed) | 0 (0.0%) | 0 (0.0%) | |
| Prefer not to say | 1 (0.4%) | 1 (1.9%) | |
| **Diagnosed health disease** | | | <.001** |
| Chronic Mental Illness | 25 (9.3%) | 4 (7.4%) | |
| Development Disability | 22 (8.2%) | 0 (0.0%) | |
| Physical Disability | 50 (18.6%) | 1 (1.9%) | |
| Neurological Disability | 18 (6.7%) | 0 (0.0%) | |
| Learning Disability | 19 (73.1%) | 1 (1.9%) | |
| Communication Disability | 20 (7.4%) | 0 (0.0%) | |
| Autism Disorder | 17 (6.3%) | 0 (0.0%) | |
| N/A | 86 (32.0%) | 45 (83.3%) | |
| Another (not listed) | 3 (1.1%) | 2 (3.7%) | |
| Prefer not to say | 9 (3.3%) | 1 (1.9%) | |
| **Financial Situation** | | | <.05* |
| Live comfortably | 147 (54.6%) | 23 (42.6%) | |
| Meet needs with a little left | 70 (26.0%) | 18 (33.3%) | |
| Just meet basic expenses | 51 (19.0%) | 11 (20.4%) | |
| Don't meet basic expenses | 1 (0.4%) | 2 (3.7%) | |

p<.05*, p<.001**.

## Association of personality with nicotine dependence in nicotine product users

As "Table 3" indicates, for all nicotine users, agreeableness was negatively associated with nicotine dependence both before ($r = -.25$, $p < 0.001$) and during COVID-19 ($r = -.19$, $p < 0.001$). Conscientiousness was also negatively associated before COVID-19 ($r = -.123$, $p < 0.001$). To examine the relationship between personality traits and nicotine dependence

**Table 2. Comparison of personality traits between nicotine users and non-users, and nicotine dependence (HONC) and perceived stress levels (PSS) among nicotine users before and during the COVID-19 pandemic.**

| Variable | Nicotine users | Non-nicotine users | P (chi-square or t-test) |
|---|---|---|---|
| **Personality** | | | |
| Extraversion | 25.73 | 23.96 | <.05* |
| Agreeableness | 29.23 | 32.85 | <.001** |
| Conscientiousness | 29.03 | 31.25 | <.001** |
| Neuroticism | 24.82 | 25.33 | .417 |
| Openness | 35.10 | 34.83 | .700 |
| **HONC (before COVID-19)** | 4.85 (sd = 3.96) | N/A | N/A |
| **HONC (during COVID-19)** | 4.71 (sd = 4.00) | N/A | N/A |
| **PSS (before COVID-19)** | 26.62 (sd = 3.63) | N/A | N/A |
| **PSS (during COVID-19)** | 26.77 (sd = 3.50) | N/A | N/A |

Personality traits were measured in both nicotine users ($n = 269$) and non-nicotine users ($n = 54$) and compared using independent-samples $t$-tests. HONC and PSS scores were only assessed among nicotine users before and during the COVID-19 pandemic.

$p < .05*$, $p < .001**$.

**Table 3. The correlation coefficients (r) and significance levels (p) between Big Five Personalities and HONC scores under three conditions: for all nicotine users (n = 269), high-stress, and low-stress groups. The stress groups are determined based on PSS scores compared to the average PSS scores before and during COVID-19.**

| Personalities | All Nicotine Users (r, p) | High-Stress Group (r, p) | Low-Stress Group (r, p) |
|---|---|---|---|
| **Before COVID-19** | | | |
| Extraversion | .096,.117 | .228, <.05* | .050,.585 |
| Agreeableness | −.245, <.001** | −.248, <.05* | −.191, <.05* |
| Conscientiousness | −.123, <.001** | −.031,.722 | −.136,.131 |
| Neuroticism | −.002,.974 | −.123,.152 | .017,.852 |
| Openness | .005,.939 | .124,.149 | −.141,.119 |
| **During COVID-19** | | | |
| Extraversion | .032,.605 | .150,.073 | .030,.757 |
| Agreeableness | −.190, <.001** | −.039,.645 | −.161,.097 |
| Conscientiousness | −.114,.061 | −.102,.223 | .014,.882 |
| Neuroticism | .034,.573 | −.031,.709 | .027,.784 |
| Openness | −.010,.873 | .052,.532 | −.005,.958 |

Independent variables: Big Five Personality traits. Dependent variable: HONC scores.

Sample sizes: High-stress ($n = 137$) and low-stress ($n = 124$) groups before COVID-19. High-stress ($n = 107$) and low-stress ($n = 144$) groups during COVID-19.

$p < .05*$, $p < .001**$.

under COVID-19 stress, participants were categorized into low and high-stress groups based on their PSS scores relative to the mean. Those scoring above the mean were placed in the high-stress group, while those below were in the low-stress group. Before COVID-19, agreeableness was negatively associated with nicotine dependence in the low-stress group ($r = −.19$, $p < 0.05$). In the high-stress group, extraversion showed a positive association with nicotine dependence ($r = .23$, $p < 0.05$), and agreeableness a negative one ($r = −.25$, $p < 0.05$). During COVID-19, no significant correlations between personality traits and nicotine dependence were observed.

"Table 4" used three regression models to examine the relationships between personality traits and nicotine dependence, personality traits and COVID-19 concerns, and COVID-19 concerns and nicotine dependence. Model one is to

**Table 4. Results of linear regression models examining the associations among Big Five personality traits, COVID-19-related concerns, and nicotine dependence (HONC scores).**

| Model | Predictor/Dependent Variable | B | S.E. | Beta | t | Sig. |
|---|---|---|---|---|---|---|
| 1[a] | **HONC During Covid** | | | | | |
| | Constant | 10.689 | 3.401 | N/A | 3.196 | <.05* |
| | Extraversion | .083 | .071 | .082 | 1.166 | .244 |
| | Agreeableness | −.170 | .064 | −.186 | −2.670 | <.05* |
| | Conscientiousness | −.062 | .066 | −.068 | −.945 | .345 |
| | Openness | .010 | .053 | .013 | .189 | .851 |
| | Neuroticism | −.021 | .062 | −.023 | −.343 | .732 |
| 2[b] | **During COVID-19 Health Concerns** | | | | | |
| | Constant | 1.686 | .319 | N/A | 5.287 | <.001** |
| | Extraversion | .008 | .007 | .084 | 1.224 | .222 |
| | Agreeableness | −.020 | .006 | −.218 | −3.213 | <.05* |
| | Conscientiousness | −.008 | .006 | −.094 | −1.357 | .176 |
| | Neuroticism | −.019 | .006 | −.218 | −3.269 | <.05* |
| | Openness | .008 | .005 | .104 | 1.548 | .123 |
| | **During COVID-19 access to Nicotine Concerns** | | | | | |
| | Constant | .959 | .300 | N/A | 3.192 | <.05* |
| | Extraversion | .019 | .006 | .204 | 3.053 | <.05* |
| | Agreeableness | −.030 | .006 | −.347 | −5.263 | <.001** |
| | Conscientiousness | −.005 | .006 | −.054 | −.798 | .425 |
| | Neuroticism | −.003 | .005 | −.034 | −.532 | .595 |
| | Openness | .007 | .005 | .107 | 1.636 | .103 |
| | **During COVID-19 economic Factors Concerns** | | | | | |
| | Constant | 1.563 | .348 | N/A | 4.489 | <.001** |
| | Extraversion | .012 | .007 | .110 | 1.618 | .107 |
| | Agreeableness | −.034 | .007 | −.335 | −5.019 | <.001** |
| | Conscientiousness | −.009 | .007 | −.091 | 1.330 | .185 |
| | Neuroticism | −.016 | .006 | −.170 | −2.596 | <.05* |
| | Openness | .003 | .005 | .039 | .583 | .560 |
| | **During Covid Mental Health Stress Concerns** | | | | | |
| | Constant | 1.632 | .315 | N/A | 5.182 | <.001** |
| | Extraversion | .007 | .007 | .075 | 1.076 | .283 |
| | Agreeableness | −.018 | .006 | −.209 | −3.024 | <.05* |
| | Conscientiousness | −.007 | .006 | −.087 | −1.224 | .222 |
| | Neuroticism | −.008 | .006 | −.093 | −1.371 | .172 |
| | Openness | .000 | .005 | .006 | .091 | .928 |
| | **During COVID-19 Routine Disruption Concerns** | | | | | |
| | Constant | 1.366 | .342 | N/A | 3.997 | <.001** |
| | Extraversion | .013 | .007 | .129 | 1.855 | .065 |
| | Agreeableness | −.023 | .007 | −.245 | −3.570 | <.001** |
| | Conscientiousness | −.003 | .007 | −.029 | −.410 | .682 |
| | Neuroticism | −.003 | .006 | −.030 | −.440 | .660 |
| | Openness | .000 | .005 | .005 | −.069 | .945 |

*(Continued)*

**Table 4.** (Continued)

| Model | Predictor/Dependent Variable | B | S.E. | Beta | t | Sig. |
|---|---|---|---|---|---|---|
| | **During Covid Social Network Concerns** | | | | | |
| | Constant | 1.431 | .289 | N/A | 4.948 | <.001** |
| | Extraversion | .005 | .006 | .060 | .857 | .392 |
| | Agreeableness | −.017 | .006 | −.215 | −3.087 | <.05* |
| | Conscientiousness | −.002 | .006 | −.024 | −.331 | .741 |
| | Neuroticism | −.006 | .005 | −.075 | −1.092 | .276 |
| | Openness | .001 | .004 | .018 | −.260 | .795 |
| | **During COVID-19 Traumatic Event Concerns** | | | | | |
| | Constant | 1.520 | .349 | N/A | 4.349 | <.001** |
| | Extraversion | .014 | .007 | .128 | 1.833 | .068 |
| | Agreeableness | −.021 | .007 | −.212 | −3.098 | <.05* |
| | Conscientiousness | −.010 | .007 | −.104 | −1.482 | .140 |
| | Neuroticism | −.010 | .006 | −.103 | −1.536 | .126 |
| | Openness | .004 | .005 | .046 | .675 | .500 |
| 3[c] | **HONC During COVID-19** | | | | | |
| | Constant | 1.383 | .713 | N/A | 1.941 | .053 |
| | Health concerns | −.469 | .797 | −.046 | −.588 | .557 |
| | Economic Factors Concerns | 2.666 | .839 | .252 | 3.180 | <.05* |
| | Mental Health Stress Concerns | 1.633 | .934 | −.180 | −1.781 | .076 |
| | Routine Disruption Concerns | 2.571 | 1.035 | .265 | 2.485 | <.05* |
| | Social Network Concerns | −.771 | .942 | −.066 | −.818 | .414 |
| | Traumatic Event Concerns | 2.139 | .942 | .226 | 2.272 | .024 |

[a] Model 1 tests the direct effect of personality traits on nicotine dependence.

[b] Model 2 evaluates how personality traits relate to various types of COVID-19-related concerns.

[c] Model 3 examines whether these COVID-19-related concerns predict nicotine dependence.

*B*: Unstandardized regression coefficient.

*S.E.*: Standard error of the coefficient.

*Beta*: Standardized regression coefficient.

*t*: t-value for the hypothesis test to determine whether the coefficient is different from zero.

*Sig.:* Significance level of the test.

*N/A*: Not applicable for the constant in the regression model.

$n = 269$ for Model 1, $n = 254$ for Model 2 and Model 3 $p < .05^*$, $p < .001^{**}$.

test the regression between personalities and nicotine dependence. Among the five personality traits, agreeableness was significantly negatively associated with nicotine dependence (B = −.170, p < .05), indicating that higher agreeableness is linked to lower nicotine dependence.

In model two, which tests the regression between personalities and COVID-19 concerns, agreeableness was consistently negatively associated with various COVID-19-related concerns, including health (B = −.020, p < .001), economic (B = −.034, p < .001), mental health stress (B = −.018, p < .05), routine disruption (B = −.023, p < .001), social network (B = −.017, p < .05), and traumatic events (B = −.021, p < .05). Extraversion was positively linked to concerns about access to nicotine (B = .019, p < .05), while neuroticism was negatively associated with health (B = −.019, p < .001) and economic concerns (B = −.016, p < .05). These findings underscore the significant roles of agreeableness, extraversion, and neuroticism in shaping responses to COVID-19-related concerns.

The regression between COVID-19 concerns and nicotine dependence was tested in model three. The regression analysis identified significant predictors of nicotine dependence during COVID-19. Access to nicotine was positively associated with increased dependence (B = 2.666, p < .05), as were routine disruption (B = 2.571, p < .05) and traumatic events (B = 2.139, p < .05). These findings suggest that these stress factors contributed to higher nicotine dependence during the pandemic. Other factors, such as health concerns (B = −.469, p = .557), economic factors (B = −1.663, p = .076), mental health stress (B = 1.204, p = .148), and social network concerns (B = −.771, p = .414), were not significant predictors of nicotine dependence.

## Discussion

This study revealed significant correlations between personality traits, stress, and nicotine dependence levels among young adults. Notably, extraversion, conscientiousness, and agreeableness showed significant differences between nicotine users and non-users. Moreover, specific stress factors related to the COVID-19 pandemic, such as access to nicotine, routine disruption, and traumatic events, were potential predictors of increased nicotine dependence. The demographic characteristics of our sample provide us with critical insights into the relationship between personal traits and smoking behavior. The first significance we observed is gender difference, with a higher percentage of males among the nicotine users, suggesting that males may have different stress-coping strategies compared to females, possibly leading to higher smoking prevalence. Another notable finding related to the health of the participants was that most nicotine users were diagnosed with a medical condition, with 69.5% diagnosed with a disease, compared to 16.7% of non-nicotine users; suggesting that individuals with physical health risks are more likely to be associated with smoking behavior. Additionally, the younger average age of non-nicotine users might reflect a generational shift in smoking behaviors due to targeted public health campaigns. Our findings are consistent with previous studies that have reported higher rates of smoking among men and older age groups "Tables 1–2". Also, we found that a higher percentage of nicotine users were bisexual compared to non-nicotine users, a novel finding that warrants further study. An explanation for this is that this population may face unique stressors or social stigma that led to higher dependence on nicotine, which is consistent with the findings on the effects of minority stress on substance use "Table 3".

Compared to non-nicotine users, active users showed higher extraversion scores and lower agreeableness and conscientiousness scores. The positive correlation between extraversion and smoking behavior is consistent with previous research suggesting that extroverts are more likely to engage in social smoking behaviors. At the same time, lower levels of agreeableness and conscientiousness among nicotine users suggest these group populations are more likely to use nicotine products, potentially due to a lack of self-regulation and empathy for others.

Regression analyses provided greater insight into the relationships among personality traits, COVID-19 worry, and nicotine dependence. Among the five personality traits, agreeableness was significantly negatively correlated with nicotine dependence (B = −.170, p < .05), indicating that higher agreeableness was associated with lower nicotine dependence. This is consistent with the negative correlation between agreeableness and nicotine dependence observed before and during COVID-19 (r = −.25, p < .001; r = −.19, p < .001) in "Table 4". The mechanisms underlying the negative association between agreeableness and nicotine dependence may be manifold. Easy-going people tend to be more cooperative, empathetic, and willing to compromise for the benefit of others. These characteristics may make them less likely to engage in behaviors that are considered harmful or socially undesirable, such as smoking. At the same time, agreeableness tends to be associated with higher levels of social support and better stress-coping strategies. During times of stress, such as the COVID-19 pandemic, easygoing individuals may be better at seeking and utilizing social support, thus reducing the likelihood that they will turn to nicotine as a coping mechanism. This hypothesis is supported by consistent negative correlations between agreeableness and nicotine dependence before and during the pandemic.

Contrary to our preconceived hypothesis, we did not find an effect on smoking behavior across personalities under the influence of COVID-19 stress from the Perceived Stress Scale. In this case, our study examined the impact of a more

comprehensive set of stressors on smoking behavior and found that agreeableness was negatively associated with all seven of our redesigned stressors, suggesting that individuals with higher levels of these traits reported fewer health worries during the epidemic. In addition, neuroticism was negatively associated with health and financial concerns as well, suggesting that these personality traits may alleviate some of the stress associated with the epidemic. However, we also found that extraversion was positively associated with concerns related to access to nicotine, which indicates extroverts felt more pressure during the epidemic from social interactions and restricted access to nicotine products. These results suggest that specific personality traits lead to altered nicotine dependence and that stress plays a moderating role.

There are several limitations to this study. First, the cross-sectional design and the sample selection limited the ability to infer causal relationships. Second, the reliance on self-reported data may be subject to recall bias and introduce social desirability bias. In addition, the sample was limited to young adults aged 18–30 years, and some specific racial groups are heavily represented, limiting the findings' generalizability to other age groups. Nonetheless, Although the Big Five Inventory is valid over time, personality is a dynamic trait. The scale can only reflect an individual's behavioral trends to a certain extent, and it has limited explanatory power for the complexity of specific behaviors. Finally, a slight adjustment of the time frame of the scale to investigate the topic of stress during the COVID-19 pandemic may have caused our study to lose certain validity.

## Conclusion

In conclusion, this study highlights the significant role that personality traits and stress play in influencing young people's dependence on nicotine. Contrary to our assumptions, however, we did not find an effect of stress on the correlation between personality and nicotine dependence within the pandemic context. Regardless of our initial hypothesis, these findings provide valuable insights for the development of targeted smoking cessation interventions and contribute to a broader understanding of smoking behavior within the context of personality psychology. Addressing specific stressors and demographic factors could improve the effectiveness of public health strategies aimed at reducing nicotine dependence.

Future studies should consider longitudinal designs to better understand the causal relationship between personality traits, stress, and nicotine dependence. Also adopting a randomized sampling approach involving samples from different age groups and cultural backgrounds would enhance the generalizability of the findings. Meanwhile, the use of some bio-indicator measures can further enhance the accuracy of the results, such as utilizing the cotinine test. In addition, exploring the relationship between personality and other substance dependencies can further the understanding of the critical nature of personal traits in addiction.

## Author contributions

**Conceptualization:** Ying Tian, Weiyi Xiang, Jessica Barrington-Trimis.

**Data curation:** Ying Tian, Weiyi Xiang, Silvia Dzhugaryan.

**Formal analysis:** Ying Tian, Weiyi Xiang.

**Funding acquisition:** Dayoung Bae, Jessica Barrington-Trimis.

**Investigation:** Ying Tian, Weiyi Xiang.

**Methodology:** Ying Tian, Weiyi Xiang, Dayoung Bae, Jessica Barrington-Trimis.

**Project administration:** Ying Tian, Weiyi Xiang, Silvia Dzhugaryan.

**Resources:** Ying Tian, Weiyi Xiang.

**Software:** Ying Tian, Weiyi Xiang.

**Supervision:** Dayoung Bae, Terry Church.

**Validation:** Ying Tian, Weiyi Xiang.

**Visualization:** Ying Tian, Weiyi Xiang.

**Writing – original draft:** Ying Tian, Weiyi Xiang, Silvia Dzhugaryan.

**Writing – review & editing:** Ying Tian, Weiyi Xiang, Silvia Dzhugaryan, Dayoung Bae, Jessica Barrington-Trimis, Terry Church.

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
