## [Decision Letter · Decision Letter 0]

PONE-D-24-54747From Traits to Puffs: The Interplay of Personality, Pandemic Stress, and Smoking BehaviorsPLOS ONE

Dear Dr. Church,

Thank you for submitting your manuscript to PLOS ONE. After careful consideration, we feel that it has merit but does not fully meet PLOS ONE’s publication criteria as it currently stands. Therefore, we invite you to submit a revised version of the manuscript that addresses the points raised during the review process. Although the study has some limitations, as highlighted in both the paper and by the reviewer, it was conducted rigorously and holds scientific validity. Please incorporate the feedback from Reviewer #2 and resubmit for further consideration.

We look forward to receiving your revised manuscript.

Kind regards,

Kamalakar Surineni, MD, MPH

Guest Editor

PLOS ONE

3. Thank you for stating the following financial disclosure:  [The Institute for Addiction Science provided Ying Tian, Weiyi Xiang, and Silvia Dzhugaryan a total of $500 to recruit and compensate participants. The grant number is not available.].  Please state what role the funders took in the study.  If the funders had no role, please state: "The funders had no role in study design, data collection and analysis, decision to publish, or preparation of the manuscript." If this statement is not correct you must amend it as needed.

4. For studies involving third-party data, we encourage authors to share any data specific to their analyses that they can legally distribute. PLOS recognizes, however, that authors may be using third-party data they do not have the rights to share. When third-party data cannot be publicly shared, authors must provide all information necessary for interested researchers to apply to gain access to the data. (https://journals.plos.org/plosone/s/data-availability#loc-acceptable-data-access-restrictions)

5. Please include a copy of Table 1,2,3,4 which you refer to in your text on page 7.

Additional Editor Comments (if provided):

Reviewers' comments:

Reviewer's Responses to Questions

**Comments to the Author**

1. Is the manuscript technically sound, and do the data support the conclusions?

Reviewer #1: Yes

Reviewer #2: Yes

2. Has the statistical analysis been performed appropriately and rigorously? 

Reviewer #1: I Don't Know

Reviewer #2: Yes

3. Have the authors made all data underlying the findings in their manuscript fully available?

Reviewer #1: Yes

Reviewer #2: Yes

4. Is the manuscript presented in an intelligible fashion and written in standard English?

Reviewer #1: Yes

Reviewer #2: Yes

5. Review Comments to the Author

Reviewer #1: The study investigates how personality traits, particularly the Big Five dimensions, and pandemic-related stressors influence nicotine dependence among young adults aged 18–30. Using validated tools like the Hooked-on Nicotine Checklist (HONC) and Perceived Stress Scale (PSS), the study finds that traits like agreeableness and conscientiousness are negatively associated with nicotine dependence, while stressors such as access to nicotine and routine disruptions during COVID-19 exacerbate dependence. Despite its strengths, including a timely focus and robust methodology, the study is limited by its cross-sectional design, reliance on self-reported data, and restricted sample diversity, which reduce the generalizability and causality of its findings. Suggestions for improvement include adopting a longitudinal design, incorporating objective biomarkers like cotinine levels, diversifying the sample to include different age groups and cultural backgrounds, and expanding the scope to stressors beyond the pandemic context. These enhancements in future studies could provide more comprehensive insights and inform targeted public health interventions.

Reviewer #2: Thank you for allowing me to review this manuscript. I enjoyed reading it and providing feedback.

Abstract:

The abstract effectively summarizes the study’s objectives, methodology, key findings, and implications. However, while it includes correlation coefficients and significance values, it would benefit from the inclusion of specific numerical data points related to nicotine dependence levels, stress scores, and demographic characteristics.

For example, adding values such as the mean Hooked-on Nicotine Checklist (HONC) scores before and during COVID-19 or specific regression coefficients from the study’s findings would enhance the clarity and immediate impact of the results.

Introduction:

The introduction provides a well-structured rationale for examining smoking behaviors in relation to personality traits during the pandemic. It successfully connects personality traits with smoking tendencies. However, the discussion of pandemic-related stress and its impact on addictive behaviors could be expanded. Incorporating recent literature on how COVID-19-related stressors, such as economic instability, social isolation, and health concerns, influence smoking habits would strengthen the theoretical foundation.

Methods/Measures:

The methods section thoroughly outlines the study’s design, including measurement instruments and statistical techniques. However, the explanation of participant recruitment and sample representativeness could be expanded. While the study mentions recruitment via social media platforms and Amazon Mechanical Turk, a discussion of the potential selection biases inherent in these methods is missing. Additionally, more details on the sample demographics, such as socioeconomic background and smoking history, would improve transparency. A discussion of how these recruitment methods may have influenced the study findings would also be beneficial.

Results:

The results section presents findings clearly and is well-supported by detailed tables. However, some portions are heavily laden with statistical information, which may be difficult for readers unfamiliar with statistical methods. Summarizing key findings in more accessible language alongside statistical results would improve readability. Furthermore, ensuring that all tables are fully explained in the narrative, particularly in terms of how the statistical relationships align with the study hypotheses, would enhance comprehension.

Discussion:

The discussion effectively ties findings back to the study’s theoretical framework and hypotheses. The analysis of how personality traits influence smoking behaviors under stress is well-structured. However, more attention could be given to unexpected or non-significant findings, such as the lack of a direct correlation between perceived stress and smoking behavior. Expanding on alternative explanations for these results, such as the role of external stressors moderating personality effects, would provide a more interpretation. Additionally, suggesting areas for future research, such as longitudinal studies examining personality and smoking behaviors over time, would strengthen the study’s impact.

6. PLOS authors have the option to publish the peer review history of their article (what does this mean? ). If published, this will include your full peer review and any attached files.

**Do you want your identity to be public for this peer review?** For information about this choice, including consent withdrawal, please see our Privacy Policy .

Reviewer #1: **Yes: ** Nikhil Tondehal

Reviewer #2: No

---

## [Author Response · Author response to Decision Letter 1]

19 May 2025

Reviewer #1: The study investigates how personality traits, particularly the Big Five dimensions, and pandemic-related stressors influence nicotine dependence among young adults aged 18–30. Using validated tools like the Hooked-on Nicotine Checklist (HONC) and Perceived Stress Scale (PSS), the study finds that traits like agreeableness and conscientiousness are negatively associated with nicotine dependence, while stressors such as access to nicotine and routine disruptions during COVID-19 exacerbate dependence. Despite its strengths, including a timely focus and robust methodology, the study is limited by its cross-sectional design, reliance on self-reported data, and restricted sample diversity, which reduce the generalizability and causality of its findings. Suggestions for improvement include adopting a longitudinal design, incorporating objective biomarkers like cotinine levels, diversifying the sample to include different age groups and cultural backgrounds, and expanding the scope to stressors beyond the pandemic context. These enhancements in future studies could provide more comprehensive insights and inform targeted public health interventions.

We have revised the Limitations section to explicitly address the limitations of cross-sectional design and self-reported data. We also added discussion on potential selection bias from MTurk and social media recruitment, and included future directions such as longitudinal designs, use of biomarkers (e.g., cotinine), broader age and cultural sampling, and stressors beyond the pandemic context.

Reviewer #2: Thank you for allowing me to review this manuscript. I enjoyed reading it and providing feedback.

Abstract:

The abstract effectively summarizes the study’s objectives, methodology, key findings, and implications. However, while it includes correlation coefficients and significance values, it would benefit from the inclusion of specific numerical data points related to nicotine dependence levels, stress scores, and demographic characteristics.

For example, adding values such as the mean Hooked-on Nicotine Checklist (HONC) scores before and during COVID-19 or specific regression coefficients from the study’s findings would enhance the clarity and immediate impact of the results.

For the abstract, we add the nicotine dependence levels, stress scores, and demographic characteristics as recommended. Specifically:

1. The total number of participants (n = 324), including 269 nicotine users and 54 non-users.

2. The average Hooked-on Nicotine Checklist (HONC) scores before COVID-19 (M = 4.85, SD = 3.96) and during COVID-19 (M = 4.71, SD = 4.00).

3. The average Perceived Stress Scale (PSS) scores before COVID-19 (M = 26.62, SD = 3.36) and during COVID-19 (M = 26.77, SD = 3.50).

Those enhance the clarity and specificity of the abstract. Thank you so much for your guidance.

Introduction:

The introduction provides a well-structured rationale for examining smoking behaviors in relation to personality traits during the pandemic. It successfully connects personality traits with smoking tendencies. However, the discussion of pandemic-related stress and its impact on addictive behaviors could be expanded. Incorporating recent literature on how COVID-19-related stressors, such as economic instability, social isolation, and health concerns, influence smoking habits would strengthen the theoretical foundation.

We have revised the introduction to include recent literature on COVID-19-related stressors, including economic instability, social isolation, and health anxiety, and discussed how these factors influenced tobacco use patterns during the pandemic. This revision provides a stronger rationale for exploring the interaction between stress and personality in relation to nicotine dependence.

Methods/Measures:

The methods section thoroughly outlines the study’s design, including measurement instruments and statistical techniques. However, the explanation of participant recruitment and sample representativeness could be expanded. While the study mentions recruitment via social media platforms and Amazon Mechanical Turk, a discussion of the potential selection biases inherent in these methods is missing. Additionally, more details on the sample demographics, such as socioeconomic background and smoking history, would improve transparency. A discussion of how these recruitment methods may have influenced the study findings would also be beneficial.

Thank you for your thoughtful feedback. We have addressed those concerns in the following aspects:

1. We specify the exclusion criteria, noting that only participants with access to internet platforms were eligible to complete the survey.

2. In the limitations section, we also expanded the discussion of recruitment methods, emphasizing the constraints of recruiting through media and Amazon Mechanical Turk and any bias it may introduce and the potential influences on the study.

3. Details regarding participants’ socioeconomic background and smoking history are available in the results section (table one).

Results:

The results section presents findings clearly and is well-supported by detailed tables. However, some portions are heavily laden with statistical information, which may be difficult for readers unfamiliar with statistical methods. Summarizing key findings in more accessible language alongside statistical results would improve readability. Furthermore, ensuring that all tables are fully explained in the narrative, particularly in terms of how the statistical relationships align with the study hypotheses, would enhance comprehension.

Thank you for your suggestions. We have revised the demographic statistics and descriptions of other results to use more accessible language and explain the results. We also updated the descriptions of the tables in the Results section to improve readability and ensure that key findings are clearly linked to the study hypotheses. Additionally, we ensured that all tables are fully explained in the narrative to enhance clarity for readers who may be less familiar with statistical methods.

Discussion:

The discussion effectively ties findings back to the study’s theoretical framework and hypotheses. The analysis of how personality traits influence smoking behaviors under stress is well-structured. However, more attention could be given to unexpected or non-significant findings, such as the lack of a direct correlation between perceived stress and smoking behavior. Expanding on alternative explanations for these results, such as the role of external stressors moderating personality effects, would provide a more interpretation. Additionally, suggesting areas for future research, such as longitudinal studies examining personality and smoking behaviors over time, would strengthen the study’s impact.

In the revised Discussion section, we explicitly address the lack of a significant correlation between perceived stress (PSS scores) and nicotine dependence. We suggest that general stress scales may not capture domain-specific or acute stressors that influence smoking behavior, especially during a unique context like the pandemic. To offer an alternative explanation, we added the possibility that external stressors—such as economic insecurity or social isolation—may moderate the relationship between personality traits and nicotine use. Furthermore, we expanded the future directions paragraph to recommend longitudinal studies, the use of refined stress measures, and exploration of moderating variables, which may clarify the complex interplay between stress, personality, and nicotine dependence.

---

## [Editor Report · Decision Letter 1]

From Traits to Puffs: The Interplay of Personality, Pandemic Stress, and Smoking Behaviors

PONE-D-24-54747R1

Dear Dr.Terry Church,

We’re pleased to inform you that your manuscript has been judged scientifically suitable for publication and will be formally accepted for publication once it meets all outstanding technical requirements.

Kind regards,

Kamalakar Surineni, MD, MPH

Guest Editor

PLOS ONE

---

## [Editor Report · Acceptance letter]

PONE-D-24-54747R1

PLOS ONE

Dear Dr. Church,

I'm pleased to inform you that your manuscript has been deemed suitable for publication in PLOS ONE. Congratulations! Your manuscript is now being handed over to our production team.

Kind regards,

on behalf of

Dr. Kamalakar Surineni

Guest Editor

PLOS ONE